# Overview of Avian Sex Reversal

**DOI:** 10.3390/ijms24098284

**Published:** 2023-05-05

**Authors:** Xiuan Zhang, Jianbo Li, Sirui Chen, Ning Yang, Jiangxia Zheng

**Affiliations:** 1Department of Animal Genetics and Breeding, College of Animal Science and Technology, China Agricultural University, Beijing 100193, China; xiuanzhang@cau.edu.cn (X.Z.);; 2National Engineering Laboratory for Animal Breeding and Key Laboratory of Animal Genetics, Breeding and Reproduction, Ministry of Agriculture and Rural Affairs, China Agricultural University, Beijing 100193, China

**Keywords:** chicken, sex determination and differentiation, sex reversal, sex control

## Abstract

Sex determination and differentiation are processes by which a bipotential gonad adopts either a testicular or ovarian cell fate, and secondary sexual characteristics adopt either male or female developmental patterns. In birds, although genetic factors control the sex determination program, sex differentiation is sensitive to hormones, which can induce sex reversal when disturbed. Although these sex-reversed birds can form phenotypes opposite to their genotypes, none can experience complete sex reversal or produce offspring under natural conditions. Promising evidence indicates that the incomplete sex reversal is associated with cell autonomous sex identity (CASI) of avian cells, which is controlled by genetic factors. However, studies cannot clearly describe the regulatory mechanism of avian CASI and sex development at present, and these factors require further exploration. In spite of this, the abundant findings of avian sex research have provided theoretical bases for the progress of gender control technologies, which are being improved through interdisciplinary co-operation and will ultimately be employed in poultry production. In this review, we provide an overview of avian sex determination and differentiation and comprehensively summarize the research progress on sex reversal in birds, especially chickens. Importantly, we describe key issues faced by applying gender control systems in poultry production and chronologically summarize the development of avian sex control methods. In conclusion, this review provides unique perspectives for avian sex studies and helps scientists develop more advanced systems for sex regulation in birds.

## 1. Introduction

Birds are important agricultural species and long-standing development model animals [1,2]. For a long time, the global poultry industry has sought effective methods to control the sex of hatchlings to achieve different purposes. In the layer industry, for example, only female chickens are required, and day-old male hatchlings are cruelly culled, which seriously increases ethical and legal concerns [3,4,5,6,7]. Deciphering the mechanisms of avian sex determination and differentiation can contribute to achieving sex control in chickens and avoiding such issues. Furthermore, owing to the high accessibility of chicken embryos, which can be genetically manipulated in vitro, they are ideal models for gonadal research [8,9,10,11]. Given the conservation of gonadal morphogenesis, studies on chicken embryos have shed light on the developmental patterns of the human reproductive system and the causes of sex disorders [12]. Thus, understanding the cell biology and genetics of gonadal formation during avian embryogenesis is crucial for improving our knowledge in diverse areas, from livestock breeding to human sex development.

In birds, the sex of offspring is determined by genetic factors. Male determinant genes activate and maintain the cascade reaction of testicular development, whereas female sex-related pathways are responsible for the proliferation and differentiation of the ovary [13,14,15]. However, avian sex differentiation is regulated by multiple factors, including genetics, epigenetics, and sex hormones [16,17]. Under natural conditions, for example, disease-induced gene mutation and alteration of estrogen levels can significantly divert the direction of sex differentiation, leading to sex reversal [17,18,19]. Although adult sex-reversed chickens form almost identical phenotypes to their opposite genetic sexes, they cannot experience complete sex reversal and produce offspring, which is caused by the cell autonomous sex identity (CASI) of avian cells [16,17,20,21]. Promising investigations indicates that hereditary factors are responsible for avian CASI. The research on the sex development of gynandromorphic birds suggests that the CASI is governed by the cellular sex chromosome combination, and is vital to the maintenance of genetic sex [21]. Until now, we have failed to clarify the underlying basis of this biological feature in birds, which needs further research.

Fundamental research on avian sex determination and differentiation can inspire the development of sex control technologies for the poultry industry, ultimately enabling the production of single-sex hatchlings. In the early years, several embryonic sex detection and selection methods based on various biological characteristics, such as egg shape, egg odors, and genetic information, were invented [22,23,24,25,26]. Although the development of these systems is limited by many challenges, such as economic benefits and animal welfare issues, they are still being applied and improved upon [27,28]. With the recent boom in CRISPR/Cas9 systems, direct editing of sex-determining genes has become possible [14,29]. This will help poultry industries to achieve precise gender regulation during production and provide insights for the study of more diversified sex control strategies.

In this review, we describe the entire process of sex determination and differentiation in birds, particularly emphasizing genetic and epigenetic regulation. Importantly, we focus on research on sex reversal in chickens and speculate that CASI might contribute to incomplete sex reversal in birds. Finally, we highlight the contribution of sex research to the advancement in gender control technologies and clarify the developmental progress of avian sex control systems. In summary, this review will help researchers understand the sex development of birds from the perspective of genetics and epigenetics, and provide theoretical support for the research of sex control methods in poultry industries.

## 2. Avian Sex Determination

### 2.1. Mechanism of Vertebrate Sex Determination

Under natural conditions, the mechanisms of vertebrate sex determination can be divided into two categories: environmental sex determination (ESD) and genetic sex determination (GSD) [30,31]. The sex-determining pattern of most reptiles, such as crocodiles, turtles, and lizards, is consistent with ESD [32,33,34,35,36,37,38]. One of the most common ESD mechanisms is temperature-dependent sex determination (TSD), which indicates that the sex of offspring is determined by ambient temperature during incubation [39]. For example, in the red-eared slider turtle, eggs of *Trachemys scripta elegans* incubated at 26 °C produce male hatchlings, whereas higher temperatures (32 °C) result in the birth of females [40]. However, GSD usually occurs in higher vertebrates, such as mammals and birds. In these species, the sex of the offspring is controlled by the combination of sex chromosomes carried by the sperm and ovum at fertilization [31]. Most mammals, such as mice, have an XX/XY sex chromosome system, which is a feature of the male heterogamety (XY) [41,42]. The Y chromosome-linked Sex-Determining Region Y (*SRY*) or male-specific repetitive DNA sequences directly govern the program of sex determination [43,44,45,46,47]. In contrast, birds have a ZZ/ZW sex chromosome system, and avian heterogametic sex is discovered in females (ZW) [48]. Studies have only identified a few groups of bona fide genes in the W chromosome, and their biological functions remain unclear [49,50,51,52].

### 2.2. Genetic and Epigenetic Regulation in Avian Sex Determination 

Previous evidence has shown that avian sex is controlled by dominant female determinant genes located on the W chromosome or governed by the Z chromosome dosage effect. The former hypothesis suggests that the W chromosome carries one or several female sex determination-related genes, which are akin to the *SRY* gene on the Y chromosome [53]. These W-linked genes are female-specific due to recombination suppression and adaptive processes, and are likely conducive to the activation of female developmental pathways [54,55,56,57,58,59]. Although the W-linked Histidine Triad Nucleotide Binding Protein (*HINTW*) and Female Expressed Transcript 1 (*FET1*) are the most promising candidates, attempts to verify their functions in ovarian development have proven unsuccessful [60,61,62,63]. Therefore, it is possible that the W chromosome may not be at the center of female sex determination. Nevertheless, the presence of the W chromosome is indispensable for maintaining female differentiation [14]. Studies have suggested that aneuploid birds with ZZW sex chromosomal karyotypes form feminized phenotypes, whereas Z0 (only possessing a single sex chromosome) develop into masculine phenotypes [64,65,66]. Hence, although current research has failed to identify one female sex-determining gene on the W chromosome, it may carry elements that stimulate female development. 

Unlike X chromosome inactivation regulated by the X Inactive Specific Transcript (*XIST*) in mammals, avian species lack global Z chromosome inactivation, which causes sex determination in avian species to be governed by the dosage effect of genes in the Z chromosome [67,68,69,70,71,72,73]. On average, male birds have two Z chromosomes; however, their female counterparts have only one. This implies that the expression level of some Z-linked genes, which are widely expressed in various tissues, is twice as high in males than in females [70,72,74,75,76]. Concerning gonads, most high-throughput studies have focused on the Doublesex and Mab-3 Related Transcription Factor 1 (*DMRT1*), a Z-specific gene that encodes a zinc-finger-like transcription factor similar to other *DMRT* family members [13,14]. The *DMRT1* gene is highly conserved across lower and higher vertebrates, and its homologs have been reported to play crucial roles in vertebrate sex development [77,78,79,80,81,82,83,84,85,86]. For instance, its duplicated copy, the DM-domain gene on the Y chromosome (*DMY*), is a promising sex determination-related gene in medaka fish (Oryzias latipes) (XX) [87,88,89,90]. As expected, this effect is also confirmed in birds. *DMRT1* has two copies in males (ZZ) but one copy in females, and has been reported to be unaffected by the dosage compensation effect [91]. The locus of *DMRT1* has been shown to generate a number of diverse alternatively spliced transcripts; however, a recent investigation suggests that only a single *DMRT1* transcript is expressed in the developing chicken gonads [92,93]. *DMRT1* is first detectable at E3.5 and is transcribed exclusively in the urogenital system [94,95,96,97,98,99]. Overexpression of *DMRT1* results in the masculinization of female embryos, characterized by the activation of testicular differentiation-related genes, SRY-box transcription factor 9 (*SOX9*), and anti-Müllerian hormone (*AMH*) [13,98,100]. Importantly, knocking out the half copy of *DMRT1* in male chickens leads to the formation of ovaries, which are characterized by the expression of female development-related markers, such as Cytochrome P450 Family 19 Subfamily A Member 1 (*CYP19A1*) and Forkhead Box L2 (*FOXL2*) [13,14,100]. This implies that two functional copies of *DMRT1* are necessary to stimulate masculinized development, and one copy of *DMRT1* is insufficient to initiate the cascade reaction of male sex determination and repress female developmental pathways in birds (Figure 1). 

Owing to the pivotal role of *DMRT1*, tight control of its spatiotemporal expression is crucial for regulating normal sex developmental programs. Evidence has confirmed that the transcriptional activity of *DMRT1* is heavily controlled by epigenetic factors, including long noncoding RNA (lncRNA) and histone modifications. For example, the Male HyperMethylated (*MHM*) region, a 2.2 kb repeat sequence located on the Z chromosome that is only transcribed in female cells from a particular strand into lncRNAs, is considered to play a crucial role in the dosage compensation and regulation of *DMRT1* expression [103,104,105]. This region is hypermethylated in males and hypomethylated in females [106,107,108,109,110,111]. The global overexpression of *MHM* inhibits the transcriptional activity of *DMRT1* in adult chicken gonads, leading to male-to-female sex reversal, suggesting that lncRNAs transcribed from this locus are likely associated with sex determination in birds [103,104,112]. In addition, current research indicates that H3K27ac, an indicator of active enhancers, is densely occupied near the *DMRT1* locus in males and is highly distributed around *MHM* in females, which is consistent with the expression pattern of *DMRT1* [113]. Therefore, based on evidences above, the absence of the dosage compensation effect highlights the importance of Z-linked *DMRT1*, which is fine-tuned by genetic and epigenetic elements, in determining avian sex.

## 3. Avian Sex Differentiation

### 3.1. Morphological Changes in Reproductive Organs during Avian Sex Differentiation 

Sex differentiation is an essential part of sex development. *DMRT1* activates sex determination as a master switch between the male and female cell fate of the gonad, and numerous genes and hormones are responsible for maintaining the subsequent process of sex differentiation underlying gonadal development in birds (Figure 1). Considering chicken as a model, gonadal progenitors appear from precursors of the coelomic epithelium and mesonephros at approximately embryonic day 2 (E2, Hamburger Hamilton Stage, HH6) [114,115,116,117,118]. The undifferentiated gonad (the so-called genital ridge), which develops on the ventromedial surface of the mesonephric kidney, primarily consists of an outer cortex and inner medulla [12,119]. Their proliferation and differentiation are mainly maintained by asymmetric cell division [9,120]. Daughter cells produced by the division remain in the coelomic epithelium and mesonephros, and the rest migrate into the genital ridge to form functional gonadal cell lineages [9,120]. At the primary stage, three types of cells are encased in the gonad: supporting cell precursors, steroidogenic progenitors, and primordial germ cells (PGCs) [121,122]. 

The supporting cell is the first bipotent somatic lineage to be differentiated under the signal of sex determination in embryonic gonads [102,123]. In chickens, embryonic mesonephros gives rise to supporting cell precursors, forming Sertoli and Leydig cells in males or pre-granulosa and theca cells in females (Figure 1) [101]. PGCs ingress into gonads, proliferating and differentiating into primary spermatocytes or oocytes, via the bloodstream from the extra-embryonic germinal crescent [124]. The morphological difference of bilateral gonads is first macroscopically visible at E5.5 (HH28) [119]. In males, the bipotential gonads develop into symmetrical testes, which are characterized by the progressive proliferation of sex cords in medullae, providing suitable niches for the differentiation of PGCs, and the flattening of outer cortical layers (Figure 1) [124,125,126]. However, the situation in females is quite different. The cortical layer of the female left gonad is well-developed, with many PGCs condensed inside, and the inner medulla appears largely unstructured, containing fluid-filled vacuoles known as lacunae (Figure 1) [127,128,129,130]. In contrast, cortical regions of the right side are only encircled by a simple epithelial layer and experience arrested proliferation in later stages [16,128,131]. 

Unlike gonads, the reproductive ducts of birds derive from two primary embryonic structures: Wolffian and Müllerian ducts [132,133,134]. In males, Wolffian ducts give rise to the vas deferens, whereas Müllerian ducts progressively degenerate under the influence of *AMH* at E9 (HH35) [135,136,137,138]. Conversely, in females, only the Müllerian duct on the left side differentiates into a functional oviduct, whereas the right Müllerian duct and bilateral Wolffian ducts gradually regress [133,139]. During ovarian formation, estrogens provide a suitable environment to prevent the left oviduct from being affected by *AMH* [140,141]. Notably, in these two seemingly independent structures, the differentiation of Müllerian ducts is closely associated with Wolffian ducts [142,143]. Ablation of Wolffian ducts in the early embryonic stages leads to the failure of Müllerian ducts [144]. Therefore, it is considered that the local signaling molecule produced by intact Wolffian ducts is required for cell proliferation and caudal migration of the Müllerian ducts [145,146]. 

### 3.2. Genetic and Epigenetic Regulation in Male Sex Differentiation 

In male birds, *SOX9*, *AMH*, and HEMOGN (*HEMGN*) are essential factors that maintain sex development (Figure 1). At the onset of gonadal differentiation, *DMRT1* acts as a pioneer transcription factor (pTF) to stimulate the downstream highly conserved genes, *SOX9* and *AMH* [13,14,98]. Chicken *SOX9* is characterized by a high mobility group (HMG) box DNA-binding domain and is upregulated during testicular differentiation [147,148,149,150,151,152]. It is mainly expressed in supporting cells and triggers the differentiation program toward pre-Sertoli cells [101,153]. The function of *SOX9* is conserved in mammals and birds. A previous study confirmed that *AMH* is the only *SOX9* target gene shared in mouse and chicken male gonads (mainly Sertoli cells) at identical embryonic periods through comparative *SOX9* chromatin immunoprecipitation sequencing (ChIP-seq) analysis [154]. This suggests that the transcriptional activity of chicken *AMH* may be regulated by *SOX9*, similar to that observed in mice. However, the expression of chicken *AMH* is first detected at E4.5 (HH25), preceding *SOX9* (E6.5, HH30) [137,155]. Therefore, *SOX9* may not directly activate *AMH* expression but may maintain it in birds. Focusing on the function of *AMH*, it is vital to the formation of avian Wolffian ducts and testes, but the overexpression or knockdown of *AMH* in embryonic chickens causes abnormal gonadal development [156]. Therefore, the proper activation of *AMH* is essential for forming the reproductive system in birds.

Another promising male sex differentiation-related gene is Z-linked *HEMGN* [157]. Research has indicated that *HEMGN* is expressed in primitive blood cells and functions in hematopoiesis in mice [158]. Avian *HEMGN*, nevertheless, is highly expressed in Sertoli cells of gonads [157]. Its expression is first detected at E5.5 (HH28), then increases significantly to a peak at E8.5 [157]. The overexpression of *HEMGN* in female embryos through retroviruses induces the upregulation of *DMRT1* and masculinization of gonads [157]. In addition, overexpression of *DMRT1* in female embryonic gonads induces the activation of *HEMGN* [98]. Hence, this evidence illustrates that a positive feedback loop exists between *DMRT1* and *HEMGN* and that the male-specific expression of *HEMGN* is pivotal to the differentiation of testes in birds.

Recently, studies found that the Transducin-Like Enhancer of Split 4 (*TLE4Z1*), which presents significant male preference at E4.5, can induce gonadal defeminization when it is overexpressed in female embryos and is involved in the masculinization of embryonic gonads [159]. Moreover, SPINDLIN1-Z (*SPIN1Z*), a Z-linked gene, is important for initiating male development [160]. Evidence has shown that overexpression of *SPIN1Z* increases the transcriptional activity of *SOX9* and *AMH* in females; however, its knockdown shows a reversed phenomenon [160]. In addition, a recent investigation pointed out that the SMAD family member 2 (*SMAD2*) is a testicular differentiation-related gene in chickens, and its disruption inhibits the expression of *DMRT1* and *SOX9* in male gonads [161]. In summary, the findings for these novel genes illustrate that sex differentiation in male birds is a complex process jointly regulated by a set of genetic factors (Figure 1).

In addition to the genetic regulatory network mentioned above, substantial evidence suggests that epigenetics is also involved in male sex differentiation. The current study revealed significant differences in transcriptome-wide m6A landscapes between E7 chicken female and male left gonads by methylated RNA immunoprecipitation sequencing (MeRIP-seq) [162]. Experiments have shown that the m6A-recognized protein YTH Domain Containing 2 (*YTHDC2*) can regulate the expression of genes related to male sex differentiation, such as *SOX9* and *HEMGN* [162]. Similarly, studies have revealed striking differences in the landscape of genomic chromatin accessibility between female and male embryonic left gonads [163]. An in-depth investigation has confirmed that the variation in chromatin accessibility corresponds to the alteration of gene expression [163]. For instance, compared to females, the transcriptional activity of *DMRT1* in males is upregulated, which is accompanied by increased chromatin accessibility [163]. Based on these findings, we conclude that epigenetic regulation plays a key role in controlling the activation and maintenance of masculine developmental pathways in birds.

### 3.3. Genetic and Epigenetic Regulation in Female Sex Differentiation 

In females, two conventional developmental pathways are considered to act in parallel to promote ovarian development. The first pathway is the *FOXL2*/aromatase (*CYP19A1*)/estrogen-signaling pathway (Figure 1). *FOXL2* is expressed in a female-specific manner and initially detected in the gonadal medulla at approximately E5.5 (HH28), just slightly ahead of the onset of sex differentiation [15,164]. In later embryonic stages, the expression of *FOXL2* also becomes detectable in a group of cortical cells [15]. The mis-expression of *FOXL2* in male gonads represses the differentiation of the Sertoli cell lineage and abolishes the local expression of testicular differentiation-related genes, such as *DMRT1*, *SOX9*, and *AMH* [15]. However, the knockdown of *FOXL2* induces ectopic activation of *SOX9* in females [15,165]. Therefore, *FOXL2* may act as a master activator of ovarian development and maintain an antagonistic relationship with male pathways. 

Downstream *CYP19A1* encodes an enzyme responsible for aromatizing androgens to form estrogens [166,167,168,169,170,171]. It exhibits a sexually dimorphic expression pattern at approximately E5.5 (HH28) and is mainly detected in pre-granulosa cells in female chickens [168,169,172]. The interference of *CYP19A1* promotes the development of masculinized medullae and inhibits the growth of cortices in female embryonic gonads, accompanied by the upregulation of *SOX9*, whereas the overexpression of *CYP19A1* induces male embryos feminization [172]. Previous studies assumed that *CYP19A1* is controlled by *FOXL2* in birds, similar to the case in mammals [173,174]. However, convincing experiments have confirmed that chicken *FOXL2* does not directly regulate *CYP19A1* [175]. The mis-expression of *FOXL2* is insufficient to stimulate *CYP19A1* in male embryonic gonads, and the disruption of *FOXL2* in females fails to influence the activation of *CYP19A1* [172]. Conversely, overexpression of *CYP19A1* increases the transcriptional activity of *FOXL2* in both sexes, and the inhibition of aromatase activity causes a reduction in *FOXL2* levels in female gonads [172,176]. Therefore, these results indicate that avian *FOXL2* is unlikely to govern the expression of *CYP19A1*, but the *CYP19A1* can control the transcriptional activity of *FOXL2*. However, it is also possible that avian *CYP19A1* is regulated by *FOXL2* in later stages [15]. 

The fine-tuning of avian *CYP19A1* is highly dependent on dynamic epigenetic modifications. In chickens, several epigenetic markers have found in the promoter region of *CYP19A1*, involving DNA methylation and histone lysine methylation [177]. These modifications exhibit different distribution patterns in male and female individuals, corresponding to the expression levels of *CYP19A1* [177]. Notably, experiments have shown that these epigenetic marks can be induced to form feminized modes in estrogen-mediated male-to-female sex-reversed chickens, indicating that DNA methylation is closely involved in governing the transcriptional activity of *CYP19A1* [177,178]. 

In birds, estrogen is an absolute requirement for female sex differentiation, as it is essential for the growth of the ovary and acquisition of secondary sexual characteristics, such as the wattle, comb, leg spurs, and feathering patterns [11,17,20,179]. Under natural conditions, estrogens stimulate the elongation and growth of Müllerian ducts, regulating the formation of tubular glands and the differentiation of the oviductal epithelium into ciliated and goblet cells [180,181,182,183]. The exogenous addition of estrogens can induce the feminization of male chickens, whereas inhibiting the production of estrogens can masculinize female chicken [16,17,91,184]. During female gonadal development, Estrogen Receptor 1 (*ESR1*) is a signal transducer of estrogens [11]. *ESR1* is asymmetrically expressed in both sexes and is restricted to cortical regions of the female left gonad at E6 (HH29) [185]. The activation of *ESR1* by propyl-pyrazole-triol (PPT) causes the formation of left-side ovotestis and retention of Müllerian ducts in male embryos [186]. Therefore, estrogens are required for female gonadal sex differentiation in birds, and their receptor *ESR1* plays an important role in ovarian development.

Another canonical pathway related to ovarian development is the Wnt Family Member 4 (*WNT4*)/ R-spondin1 (*RSPO1*)/*β-catenin* signaling pathway (Figure 1). *WNT* morphogens are a highly conserved family of signaling molecules that play a crucial role in the differentiation of female gonads in several animals [12,187,188]. The binding of *WNT4* with its coreceptor LDL Receptor Related Protein 6 (*LRP6*) and *RSPO1* allows the formation of a complex that can promote the expression of *β-catenin* by inhibiting its phosphorylation [189,190,191]. In chickens, *WNT4* is expressed in bipotential gonads of both sexes at E4.5 (HH25) [192]. Its expression level gradually increases in female gonads from E6.5 (HH30) to E8.5, which is mainly restricted to pre-granulosa cells in the left cortex and decreases in males [192]. Chicken *RSPO1* shows a sexually dimorphic pattern from early stages. Its expression is detectable in female embryonic gonads as early as E4.5 (HH25) and increases strongly from E8.5 [192]. *RSPO1* is co-expressed with *WNT4* in cortical regions of the female left gonad and loses its expression on the right side since E8.5 [192]. As for *β-catenin*, its expression is detected in the female left gonad at E6.5 (HH30) and increases significantly in cortical regions at E13.5 [193]. Our knowledge of this pathway derives mainly from mammals and humans. Mutation of the *WNT4* gene in humans causes female-to-male sex reversal, which is coupled with renal, adrenal, and lung dysgenesis [194]. Similarly, the lack of *WNT4* and *RSPO1* in mice induces the differentiation of ovotestes in females and the maldevelopment of testes in males [195,196]. Moreover, knocking out *RSPO1* in female mice results in decreased levels of *β-catenin* signaling molecules and female-to-male sex reversal [197]. However, in birds, we can only speculate that the function of *β*-catenin is identical to that in mammals. The amino acid sequence of the *β*-catenin gene is highly conserved in vertebrates, with 99% similarity between chicken and mice [198]. Therefore, it is likely that *β*-catenin plays a conserved role in chicken ovarian development and female sex differentiation, which needs to be proven in future studies.

Although these two pathways related to ovarian development are expressed in different gonadal regions, *FOXL2* and *CYP19A1* are expressed mainly in the medulla, whereas the *WNT4*/*RSPO1*/*β-catenin* pathway is detected in the cortex. It has been shown that there are some interplays between them [193]. An investigation found that inhibiting the synthesis of estrogens through aromatase inhibitors curbs the expression of *RSPO1* in embryonic ovaries [192]. However, it is unclear whether this effect is caused by the direct interaction of estrogens with *RSPO1* gene transcription or the lack of well-developed cortical regions in sex-reversed gonads, which results in leaving a few groups of cells to express *RSPO1*.

Several genes, independent of these pathways, have recently been identified to play crucial roles in ovarian development (Figure 1). Experiments have shown that the Jun proto-oncogene, AP-1 transcription factor subunit (*JUN*) functions as an ovarian differentiation-related regulator in embryonic periods [161]. Overexpression of *JUN* inhibits the expression of *SMAD2*, *DMRT1*, and *SOX9* in male embryonic gonads while inducing the expression of *FOXL2*, *ESR1*, and *CYP19A1* [161]. However, the knockdown of *JUN* leads to the masculinization of female embryonic gonads [161]. A recent study identified a novel regulator of juxtacortical medulla differentiation in female embryonic gonads, termed TGF-β Induced Factor Homeobox 1 (*TGIF1*) [199]. It is mainly expressed in the cortex and the pre-granulosa cell lineage [199]. Although estrogen-mediated male-to-female sex reversal can induce ectopic activation, results from targeted mis-expression and knockdown of *TGIF1* indicate that it is only required, but insufficient, for proper ovarian cortex formation [199]. These results suggest that ovarian differentiation is a polygenic regulatory process. Two signaling pathways, *FOXL2*/*CYP19A1*/estrogen, and *WNT4*/*RSPO1*/*β-catenin*, play significant roles in the developmental program.

## 4. Avian Sex Reversal

### 4.1. Occurrence of Vertebrate Sex Reversal 

In birds and some lower vertebrates, sex differentiation is often affected by various factors, leading to sex reversal. In chicken flocks, the population’s social structure can influence the sex of individuals, similar to the case of fish [200,201,202,203,204]. For instance, when there is no rooster in a chicken flock, a hen may experience sex reversal to maintain the reproduction of the population [135]. This is an adaptive change that arises during the long-term evolutionary process. However, no further research has been conducted to prove this phenomenon. In addition, diseases are a major cause of sex reversal. Ectopic activation of aromatase and accumulation of estrogens as a result of the henny-feathering trait, an autosomal dominant mutation, can feminize the feathering patterns of male chicken [18,205]. However, this disease-mediated sex reversal might not influence gonadal morphology because aromatase is mainly mis-expressed in extragonadal tissues [18,205].

### 4.2. Mechanism of Avian Sex Reversal 

Bulk instances of sex reversal in birds are associated with the high sensitivity to alterations in sex hormone levels [16,206]. This biological characteristic of birds is closely related to their evolutionary processes. From the viewpoint of evolution, mechanisms of mammalian sex determination and differentiation evolved from synapsid reptiles, whereas diapsid reptiles gave rise to crocodiles, lizards, and birds 150 million years ago [125,207]. In eutherian mammals, gonadal sex differentiation appears independent of sex steroid hormones and can proceed without steroidogenesis [208,209]. Environmentally insensitive gonads of highly evolved mammalian embryos occur in the maternal womb, which is a potentially dangerous place rich in various hormones (Figure 2) [210]. Hence, the highly evolved placenta and maternal internal pregnancy pattern may have forced eutherians to abandon estrogens as components of the sex-determining cascade. However, sex differentiation of birds occurs in an extra-maternal environment and has characteristics similar to those of lower vertebrates (Figure 2). Therefore, it is likely that they do not have the mechanism to resist the alteration of exogenous estrogen levels, which is usually unlikely to occur under normal hatching conditions. Therefore, the evolutionary position of birds and the biological characteristics of extra-maternal embryogenesis indicate that avian sex differentiation is prone to being affected by environmental hormones.

### 4.3. Avian Sex Reversal Induced by Transplant Treatment 

Sex-reversed birds can be obtained by altering endogenous hormone secretion through gonadal grafting. Transplantation of E13 (HH39) chicken whole testes into E3 female chicken extra-embryonic coelom can induce gonadal sex reversal [211,212]. Under the stimulation of exogenous testes, the left gonad differentiates into a testis instead of an ovary, and germ cells migrate into sex cords. However, they do not experience a meiotic process [212]. Moreover, grafting E2 female (male) sections of the presumptive mesoderm, which gives rise to gonads, to replace the equivalent tissue of male (female) at the same growth stage, induces the formation of mixed-sex chimeras [21]. Therefore, artificial transplantation of exogenous tissues can affect avian sex differentiation, leading to sex reversal.

### 4.4. Avian Male-to-Female Sex Reversal Induced by Estrogens Treatment 

In most cases, sex-reversed birds are created by adding estrogens or aromatase inhibitors to manipulate endogenous hormone levels (Figure 3) [16,17]. The administration of estrogen to quail eggs during the first half of embryonic life (before the onset of sex differentiation) can feminize genetic males, characterized by the hyperproliferation of the left gonadal cortex and vacuolized inner medulla [16]. Moreover, injecting a 17a-Ethinylestradiol (EE2) emulsion into E3 quail eggs can demasculinize male individuals, which will lose typical masculine sexual behavior and form asymmetric testes with decreased areas of androgen-dependent cloacal glands after maturity [213]. Recently, research has found that treating chicken eggs with this emulsion can feminize male chickens to different degrees [177,178]. The left gonads of these individuals progressively form female-like cortical regions and fluid-filled medullae from low to high degrees of sex reversal, while the characterized testicular tissues are almost lost [178]. However, the effect of estrogen-mediated male-to-female sex reversal is transient. Some reversed individuals revert to normal male phenotypes after hatching, whereas others can persist in female phenotypes for not more than 1 year [125]. In addition, estrogen-regulated sex reversal is not limited to gonads but is also presented in secondary sexual characteristics. Injection of estradiol into leg muscles of adult male chickens can feminize feathering patterns, including reducing saddle feather length, increasing plumulaceous segments, and altering feathering colors (Figure 3) [214]. Similarly, these changes are not permanent and gradually disappear with decreased estrogen levels in the body. Taken together, the addition of estrogens can significantly affect the differentiation of gonadal structures during embryonic periods and influence the formation of secondary sexual characteristics in adult periods; however, these effects are only maintained for a short time. 

### 4.5. Avian Female-to-Male Sex Reversal Induced by Aromatase Inhibitors Treatment 

Injecting aromatase inhibitors, which reduce the production of gonadal estrogens, into avian eggs can induce female-to-male sex reversal (Figure 3) [17,215,216,217]. A previous study found that treating E3 chicken eggs with fadrozole, an aromatase inhibitor, can masculinize bilateral female embryonic gonads, which fail to form stratified cortexes, but develop functional medullae enclosing germ cells at E9.5 [218]. Notably, aromatase inhibitors-treated chickens exhibit varying degrees of sex reversal (Figure 3) [184]. The gonadal cortex is reduced in highly sex-reversed individuals, and germ cells are relocated into developed medullary structures [184]. During the development of sex-reversed female embryos, *DMRT1*, *SOX9*, and *AMH* expression is significantly upregulated in gonadal medullae, whereas *FOXL2* and *CPY19A1* are downregulated in gonadal cortexes with fewer residues in juxtacortical medullae [91,218,219]. In addition, the primary AMH Receptor, AMH receptor type-II (*AMHR2*), is upregulated in fadrozole-treated females and co-locates with *DMRT1* in Sertoli cells [220]. Moreover, gonadal morphology and inner structures of sex-reversed chickens continue to change after hatching. In 1-day-old sex-reversed chickens (D1), both follicles and tubular structures are visible in the gonads [221]. These tubular structures differentiate into abnormal seminiferous tubules at D11 and form atypical tubules with areas of loose connective tissue at D21 [221]. The left and right gonads grow into small ovotestes in D42 chickens, containing greatly enlarged atypical seminiferous tubules and fewer normal appearing seminiferous tubules [221]. This consecutive alteration is due to the permanent effect of aromatase inhibitors on chicken sex differentiation, which is likely achieved by cell reprogramming during the embryonic period. Research has emphasized that fadrozole-mediated female-to-male sex reversal includes a crucial event, namely, pre-granulosa cells trans-differentiate into undifferentiated *PAX2^+^* supporting cells before forming Sertoli cells, which is lost in estrogens-mediated male-to-female sex reversal [222]. However, current evidences are still unable to clarify the potential mechanism of this phenomenon, which needs to be studied in detail. 

The permanent effect of aromatase inhibitors on avian sex differentiation can significantly influence the development of reproductive tissues and the appearance of birds. Adult sex-reversed chickens form almost symmetrical gonads and small testes with regressed oviducts on the left side [17,107,223]. In addition, injecting aromatase inhibitors before sex differentiation during the embryonic period or after sexual maturity in the adult period can induce testosterone production while inhibiting estradiol synthesis in the blood [223,224]. The sex-revered females progressively form masculinized hackles, saddle feathers, combs, and wattles. In contrast, their body weights, especially muscle mass and fat pad weight, remain in line with normal females, suggesting that growth performance, unlike secondary sexual characteristics, may be controlled directly by genetic factors and independent of hormones [214,221,223,224,225]. Notably, under the influence of an aromatase inhibitor, Wolffian ducts in sex-reversed chickens develop into vas deferens, and gonads can produce sperms carrying the W chromosome [226]. Further investigation has shown that W-carrying sperms have oocyte-activating potency and can induce the formation of male and female pronuclei [227,228]. In summary, aromatase inhibitors can significantly disrupt estrogen synthesis and permanently influence the process of sex differentiation, causing the formation of sperm-carrying small testes and masculinized secondary sexual characteristics in adult sex-reversed birds.

### 4.6. Avian Sex Reversal and Cell Autonomous Sex Identity 

Sex reversal can be induced in birds, but neither does the male bird has the ability to form well-functional ovaries, nor do female birds show normal masculinized courtship behavior and successfully produce fertile sperm after sexual maturity [17,229]. This incomplete sex reversal may be regulated by the CASI of avian somatic cells, which is a cellular inherent sex identity that can help individuals maintain their genetic sex independently of the impact of hormones [21,74]. Current research points out that the CASI is determined by the genetic information, especially that carried by sex chromosomes, and epigenetic factors, such as DNA methylation and histone lysine methylation [21,177]. At early developmental stages when primary gonads have not been induced to differentiation, sexually dimorphic gene expression can be detected in several tissues between male and female birds, indicating that the CASI is widely involved in organogenesis programs [230,231]. Studies of the CASI mainly focus on gynandromorphic birds, which display a significant bilateral asymmetry; one side of the body appears to be male, but the other appears to be female [74]. In gynandromorphic chickens, most cells on the side with a female appearance are of the ZW genotype, and most cells on the other side with a male appearance are of the ZZ genotype [21]. Notably, the gonadal morphologies of these gynandromorphic chickens conform to their cellular compositions, testicular and ovarian appearances are mainly composed of ZZ and ZW genotypic cells, whereas ovotestis is composed of a mixture of both cells [21]. Consistently, male-determining genes, such as *DMRT1* and *SOX9*, are highly expressed in Sertoli cells of the seminiferous cords on the masculinized side, where aromatase is not detectable [232]. However, the gonads of the feminized side show areas of peripheral aromatase expression, together with areas of *SOX9*-positive and *DMRT1*-positive seminiferous tubules [232]. Moreover, the concept of CASI has been confirmed in mixed-sex chicken chimeras. Cells from donor females are restricted to the interstitial tissue between the sex cords and are not recruited into functional Sertoli cells in testes. Similarly, donor male cells cannot be recruited to form granulosa cells in ovaries [21]. That is, there is no interaction between *GFP*+ donor female cells and *AMH*+ host male cells in the testes, and *GFP*+ donor male cells and *CYP19A1*+ host female cells in the ovaries, suggesting that the differentiation of these cells is confirmed by their genetic fates [21]. Notably, epigenetic factors are likely to contribute to maintaining gonadal fate. The chromatin accessibility of *DMRT1* in embryonic female-to-male sex-reversed chickens cannot be increased to the normal male level, corresponding to its transcriptional activity [163]. In addition, the pattern of DNA methylation around *CYP19A1* in estrogen-mediated sex-reversed chicken gonads fails to be induced to form the same as that in control females, suggesting that epigenetic elements are involved in the establishment of avian gonadal CASI [177,178]. 

In addition, studies have found that the CASI also affects the growth of avian secondary sexual characteristics [233,234]. In a gynandromorphic finch, although both halves of the brain are exposed to the same hormone environment, the neural song circuit of the male side has a more masculine phenotype than that of the female side [76]. Strikingly, recent research found that *DMRT1*-knockout male chicken, which had ovaries instead of testes, can form the normal masculinized appearance, featuring large combs and wattles, hackle feathers, and long leg spurs in adult periods [14]. Similarly, in gynandromorphic chickens, the female side forms sex-linked feathering patterns, smaller spurs, and wattles than those on the male side after sexual maturity [232]. These results suggest that avian CASI may be critical in differentiating sex phenotypes. 

## 5. Avian Sex Control

### 5.1. Critical Issues and Challenges of Avian Sex Control 

One of the most important purposes of avian sex research is to implement it in the poultry industry to solve practical production problems [235,236,237]. At present, billions of 1-day-old male hatchlings are culled globally every year, and their carcasses are further processed into feed by machines or directly buried into soils [3]. Although those methods save the feeding cost, they severely violate animal welfare regulations advocated by a number of countries, and untreated animal carcasses are very likely to cause biosafety hazards which can endanger human health [238,239]. Inspired by gender control systems applied in dairy farms, the development of avian sex control technologies is a promising means to solve the abovementioned concerns and improve breeding efficiency [240,241]. Investigations of avian sex determination, sex differentiation, and sex reversal have provided abundant insights into the development of early sex detection and sex control technologies [14,101,242]. However, sex research in birds is a huge puzzle, full of various unknown regulatory networks, and we cannot clearly explain the detailed molecular biological mechanisms of sex determination and differentiation in birds at present. This creates huge barriers to the progress of avian gender control systems. 

In addition, the assumption of the application process in poultry industries greatly limits the research direction and fields of sex control technologies. To ensure the economic benefits of the poultry industry, the applied technologies cannot affect the hatchability of eggs and the growth of chicken embryos, nor can they influence the egg production and survival rate of laying hens [27,243,244]. As a successful gender control strategy must help improve production efficiency, the system should be simply operated and highly mechanized, which can be used in intensive breeding [245,246]. Moreover, biosafety, food security, and animal welfare must be considered in the research process of gender control strategies. Those methods must fit into the political environment and be acceptable from a humanitarian and ethical point of view [28,247,248,249]. For example, sex control technology must be applied before chicken embryos’ pain perception has evolved. The initial sensory afferent nerves develop in the chicken embryo on E4 of incubation, but a synaptic connection to the spinal cord is not present before E7 of incubation, which makes nociception impossible in the first third of incubation, suggesting that the intervention of embryonic sex development must be earlier than E7 in chickens [250,251]. In summary, the development of sex control approaches has been, and is currently, still being measured according to the abovementioned criteria, with many challenges to be overcome on the way from an idea to practical application. 

### 5.2. Research Progress and Application of Avian Sex Control Technologies 

With the in-depth development of basic research and interdisciplinary communication, several gender control technologies have been developed, and some systems have even been used in poultry production [23,252,253]. In the early years, the focus of research was, for instance, to investigate the connection between the outer shape of the eggshell and the sex of hatchlings, as well as the association between the egg odor and the embryonic gender (Figure 4) [25,26]. Those studies have provided pivotal perspectives for developing avian sex control systems. However, there is no corresponding data on the likelihood of in ovo sex control based on egg shape and odor attributes [254].

In addition to using morphometric and chemical signals as biomarkers for sex detection, different optical and imaging methods have been successfully performed in birds. A great advantage of optical methods is their contactless nature. Raman spectroscopy, a representative subfield of vibrational spectroscopy based on the so-called “Raman effect”, has been widely employed in biological detection and clinical research [255,256,257]. It is unique to each molecule and is often called “molecular fingerprints.” As the biochemical composition of female and male cells is significantly different, Raman spectroscopy allows for sex identification in ovo according to the spectral signature of germinal or blood cells (Figure 4) [258,259]. For example, sex-specific molecules obtained from the E3 or E5 developing chicken vascular system when embryos have the ability to resist the impact of external stimuli and are earlier than the connection between afferent nerves and spinal cord can be used to distinguish between male and female eggs, respectively [259]. In addition to Raman spectroscopy, some promising spectra, such as Fourier transform infrared spectroscopy and time-resolved laser-induced fluorescence spectroscopy, are also being tested and applied to the sex control of poultry industries [253,260]. With the advance of these optical systems, the Canadian company “Egg Farmers of Ontario” funded a new technology called “Hypereye” which uses hyperspectral imaging to identify infertility and gender of day-of-lay eggs. As those eggs are essentially the same as regular table eggs, early identification could indicate a new source of eggs, including but not limited to the production of human vaccines and anti-depressants [261,262]. Moreover, the US company “Vital Farms” and Israeli company “Novatrans” jointly developed a new technology named “TeraEgg”. This system uses infrared light combined with complex algorithms to analyze the signs of sex and reproduction from the early stage of the embryonic development of chicken eggs. This implies that farms and intensive breeding industries can distinguish between male and infertile eggs, which can be brought back to the market and sold to the public. Multiple systems, including the two instructive sex control methods mentioned above, are being consistently improved with the research progress of avian sex development and spectroscopy and will finally be applied in the poultry industry.

In addition to achieving sex control using optical and imaging methods, the thriving genetically engineered technology represented by the CRISPR/Cas9 system provides a direct way to govern the gender of embryonic chickens. As genes regulate the determination of avian sex, the genetic marking of sex chromosomes has also been discussed as a possible route for in ovo sex control in birds (Figure 4) [14,21]. Based on previous knowledge, several studies have focused on the production of genetically engineered hens and have described the marking of the Z chromosome of breeding hens with various fluorescent proteins [3,263]. By labeling the Z chromosome of breeding hens with fluorescent biomarkers, when mated to non-genetically modified roosters, male-determined eggs can be identified based on their fluorescence signals before incubation or at early-stage incubation. Notably, this method was successfully used for sex control in layers, with sex being detected from sex-specific patterns of germinal disc fluorescence in non-incubated eggs [264]. However, according to current knowledge, this strategy cannot be operated once and for all, because female chickens hatched from such mating do not show any genetically modified genes. 

Furthermore, based on gene-editing technologies, research has found that environmental factors can also activate or inhibit the expression of sex-determining genes, thus achieving avian sex control [265,266]. The US company “ONCE” has developed a “genetic switch” that can be used to selectively activate or deactivate light absorption centers in sex-determining genes using narrowband light-emitting diodes. In this way, the sex of oviparous embryos can be influenced at early differentiation stages. It should be possible to selectively produce only offspring of the desired sex (females in laying hen production and males in broiler production). Similarly, the Israeli company “Soos Technology” has invented a solution to affect the gene expression of the reproductive system in genetic male embryos, which can turn males into functional females capable of laying eggs. The artificial intelligence-guided incubation system developed by this company uses a patented combination of temperature, humidity, CO_2_, and sonic vibrations to control the sex development process in poultry embryos and turn genetically male chickens into functional females. However, no reliable results have been published, and no information is available concerning trials or the practical use of those methods. Nevertheless, those non-contact gene expression regulation modes provide new ideas for follow-up research on gender control technologies and are expected to open up more advanced sex control strategies in the near future.

## 6. Conclusions and Future Perspectives

The sex development of birds is a precise process. Research has indicated that the dosage effect of *DMRT1* is the master switch that controls the activation of avian sex determination [13,14]. However, subsequent sex differentiation is a complex process that involves multiple genetic factors. To date, studies have identified several important genes related to this program, e.g., *SOX9*, *AMH*, and *HEMGN* in males and *FOXL2*, *CYP19A1*, *WNT4*, and *RSPO1* in females, but have failed to elaborate the regulatory network between these genes and the molecular basis of the everlasting antagonistic relationship between testicular and ovarian pathways [13,15,187]. Therefore, future studies are needed to explore the internal regulatory mechanisms between avian sex-related genes.

In birds, sex reversal can be induced by various methods, including but not limited to the addition of exogenous estrogens and aromatase inhibitors. Sex-reversed birds are important for studying avian sex determination and differentiation because of their unique sex development modes [17,20,176]. Notably, regardless of the type of treatment used, the chicken cannot achieve complete sex reversal, even during embryonic periods. Current evidence indicates that this is due to the CASI of avian somatic cells [21]. The alteration of hormone levels can only cause partial or temporary sex reversal in birds. The original CASI of avian cells is the main force to resist external stimuli during the entire developmental stage, a biological characteristic that evolved over a long period of time [21,74]. However, the molecular mechanism of avian CASI still remains unclear, thus warranting further studies.

In recent years, sex research in mammals has suggested that epigenetic regulation plays a pivotal role in governing the expression of sex-related genes [267,268]. Epigenetic marks are reversible modifications that show varying patterns between males and females before and after sex development [269,270,271]. Previous studies have shown that several epigenetic elements are involved in sex development, such as lncRNA, DNA methylation, histone modification, and RNA methylation [107,162,177]. Current research has enabled the establishment of a regulatory relationship between epigenetics and avian sex determination and differentiation. However, the regulatory mechanism of epigenetic modifications on the activation of sex-related genes remains unclear. With the development of new technologies, such as high-throughput chromosome conformation capture (Hi-C), researchers can build three-dimensional chromatin landscapes at different time points of avian sex development and identify sex-related enhancer hubs [272,273]. Those results will help us investigate the 3D regulatory landscape of sex determination and the molecular basis of the transition from an initially biopotential gonad to either alternative fate in birds. Therefore, future sex research should pay more attention to the influence of 3D chromatin interactions on gene expression and look for epigenetic modifications that truly regulate sex-related gene expression in birds.

The ultimate goal of both genetic and epigenetic research in avian sex development is to employ it in the poultry industry and help solve problems, such as the uncertainty of sex-determining factors and barriers to the development of ideal systems, related to sex control [235,236,237]. Regarding gender control technologies, the most important thing is to find a “source of information” that provides reliable information about the sex of embryos as early as possible and cooperate with machines to identify single-sex fertilized eggs. Biomarkers employed by avian sex control have gradually improved from eggshell features and egg odors to biological signals revealed by spectroscopic technologies [25,26,258,259]. Advancements in this process emphasizes the importance of basic research and interdisciplinary communication. In the future, with the development of spectroscopic technologies and the reduction in machine costs, it will be possible to introduce spectroscopic detection equipment into laying hen hatcheries on a large scale and even introduce magnetic resonance imaging technology for early sex detection [22,274]. In addition, the direct impact of gene-editing systems and environmental stimuli on activating and suppressing sex-determining genes provides new perspectives for sex regulation in the poultry industry. Those non-invasive and efficient methods reduce the cost of gender detection and truly achieve one-step gender control in poultry production. However, whether the gender-control technologies that affect gene expression will cause biosafety problems and how to address corresponding ethical issues still need further discussion. 

## Figures and Tables

**Figure 1 ijms-24-08284-f001:**
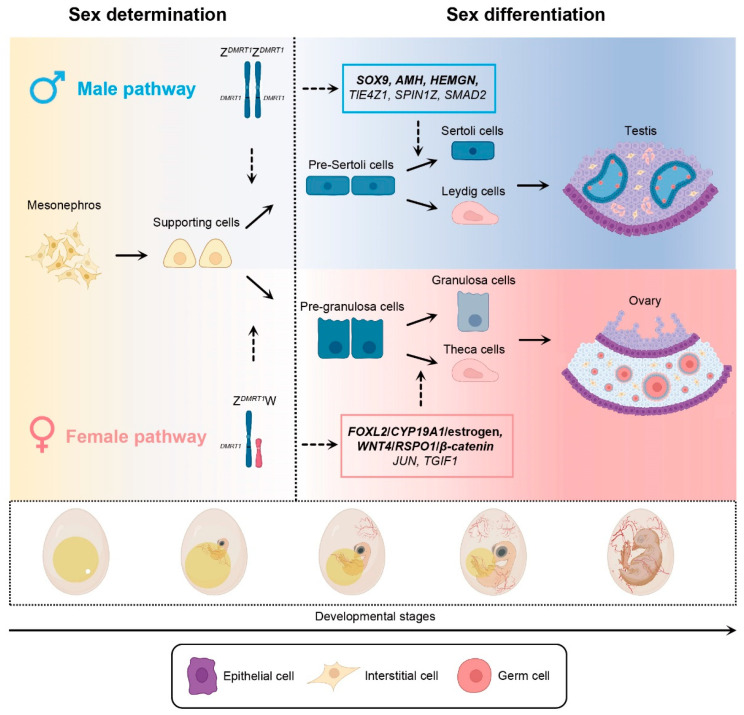
Overview of the derivation of chicken supporting cells and key genes promoting bipotential supporting cells to adopt either testicular or ovarian development during embryonic stages. The mesonephros gives rise to supporting cells which further form Sertoli and Leydig cells in males or granulose and theca cells in females. In embryonic ZZ gonads, two copies of *DMRT1* trigger the differentiation program from supporting cells to pre-Sertoli cells, and several downstream male-determining genes further induce pre-Sertoli cells to form Sertoli and Leydig cells. In the embryonic ZW gonads, one copy of *DMRT1* cannot activate the masculine development, and thus female supporting cells differentiate into pre-granulosa cells, which are further induced to form granulosa and theca cells under feminized signals. Modified from Estermann’s articles with permission [101,102].

**Figure 2 ijms-24-08284-f002:**
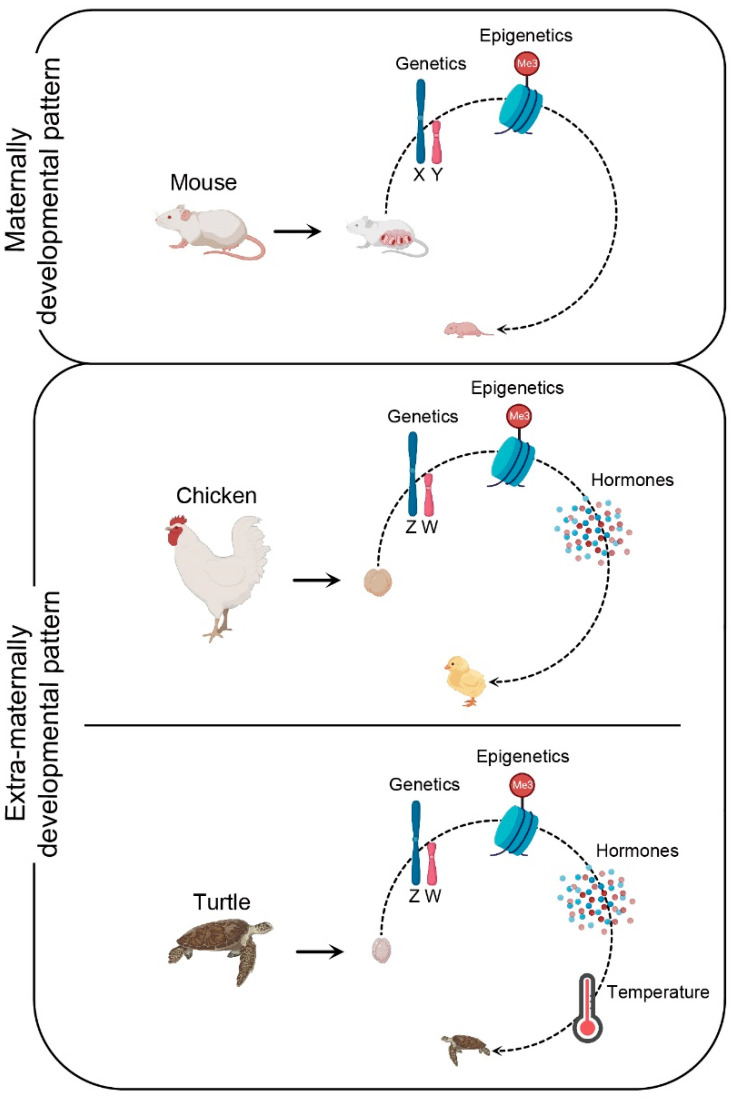
Schematic view of key factors influencing the embryonic sex determination and differentiation in maternal and extra-maternal developmental patterns. In mouse, the process of embryonic sex development occurs in the maternal environment, and is mainly regulated by genetic and epigenetic factors. In chicken, the process of embryonic sex development occurs in the extra-maternal environment, and is governed by genetic and epigenetic factors and sex hormones. Likewise, the process of turtle embryonic sex development occurs in the extra-maternal environment, and is controlled by genetic and epigenetic factors, sex hormones, and temperature.

**Figure 3 ijms-24-08284-f003:**
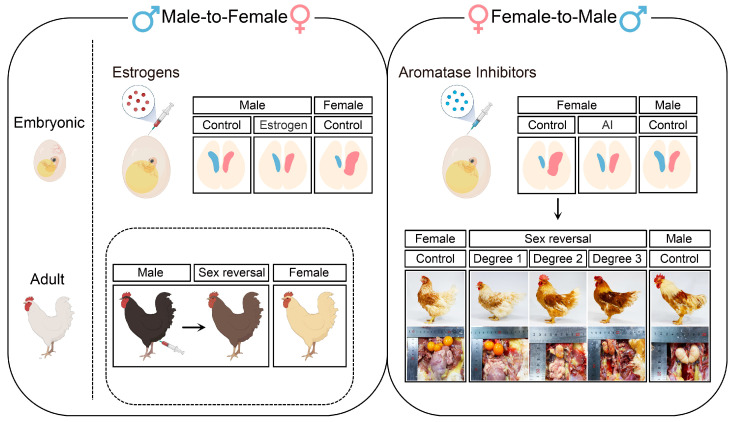
Overview of gonadal phenotypes and secondary sexual characteristics in embryonic and adult sex-reversed chickens created by treatment with either estrogens or aromatase inhibitors. The left panel: Injection of estrogens into chicken eggs can lead to male-to-female sex reversal, characterized by the proliferation of the left gonad and degeneration of the right gonad. During adult period, injection of estrogens into leg muscles of male chicken can induce the feminization of feathering patterns. The right panel: Injection of aromatase inhibitors into chicken eggs can lead to female-to-male sex reversal, characterized by the symmetrical proliferation and differentiation of bilateral gonads. During the adult period, the secondary sexual characteristics and gonadal appearance show different degrees of sex reversal in aromatase inhibitors-treated female chicken.

**Figure 4 ijms-24-08284-f004:**
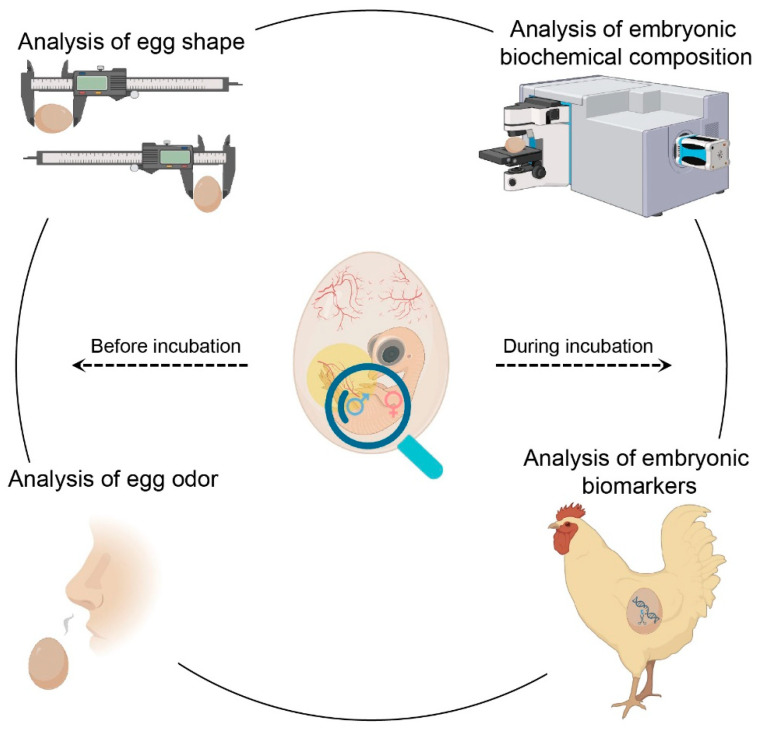
Schematic view of early avian sex control technologies. Before incubation, previous studies attempted to clarify the association between the egg shape and the sex of hatchlings, as well as the correlation between the egg odor and the embryonic gender. During incubation, research focused on identifying the sex-specific biochemical composition by spectroscopic detection and investigating fluorescent biomarkers created by gene-editing systems to detect the gender of embryos.

## Data Availability

Not applicable.

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
