# Peer review of "Overview of Avian Sex Reversal"

_ijms, 2023, doi:10.3390/ijms24098284_

Round 1

Reviewer 1 Report

The authors have submitted a review article that outlines current knowledge and research gaps regarding avian sex determination and differentiation as well as their sex reversal. The review of current scientific literature is of great importance from the standpoint of animal welfare, public, scientists and poultry production.

The title, abstract and keywords accurately reflect the content of the manuscript. The authors of this manuscript gave us a clear introduction based on the currently available scientific literature in this filed. Other sections in the manuscript are clearly defined and well discussed. In the Conclusion section authors summarised the main points regarding avian sex determination, sex differentiation r and sex reversal and emphasised research gaps that have to be investigated in the future studies. References consist of appropriate and relevant papers.

According to my opinion, the manuscript should be accepted for publication in International Journal of Molecular Sciences with only a few minor corrections (as outlined in the submitted PDF file).

Reviewer 2 Report

The tropic of this manuscript is interesting and well-motivated. However, for further improvement, I have the following comments.

1. In order to make this review easier for readers from different research backgrounds to understand, please provide a brief description of avian Cell Autonomous Sex Identity (CASI) in 4.6. Avian Sex Reversal and Cell Autonomous Sex Identity.

2. Please provide some description about the practical significance of avian sex control technologies for poultry production in 5.1. Critical Issues and Challenges of Avian Sex Control.

Reviewer 3 Report

Dear editor in chief and authors,

This study aims to review the current data regarding avian sex reversal, from the avian sex determination, avian sex differentiation to avian sex reversal and avian sex control.

The authors has realized a very complex review, very well documented and written in proper English. The study is interesting and can be accepted for publication in the present form.

I read the manuscript with great interest. The manuscript is well written and structured, clear and straight-forward to understand

Thank you!
